# Rate/Distortion Constrained Model Quantization for Efficient Storage and Inference

## Abstract

The proliferation of large pre-trained neural networks has recently revived research in both quantization of network weights (for faster inference), and in their compression (to reduce file sizes). However, there has so far been little idea transfer between the two lines of research. In this paper, we combine techniques from quantization and compression to propose an efficient and highly effective post-training compression method for large neural networks. Our method extends the recently published quantization method OPTQ (Frantar et al., 2023) with a tunable rate/distortion trade-off by introducing a cost per bit into OPTQ's rounding operation. Crucially, we estimate the bit rate based on the predictive model used in the state-of-the-art neural network compression method NNCodec (Becking et al., 2023). In our experiments with several standard pre-trained networks from the computer vision community, our method leads to significantly (up to 2.7x) smaller file sizes than NNCodec at equal model performance, generally compressing to less than half a bit per network weight and implicitly pruning insignificant weights. Additionally, and in contrast to NNCodec, our method offers the same opportunities for inference speed-ups as OPTQ. By proving that file size and inference cost can be reduced simultaneously, we hope that our contribution shows a path towards deploying large neural networks on end-user devices, alleviating privacy concerns, regulatory constraints, and dependency on large service providers.

## 1 Introduction

Neural networks have achieved impressive performances in a large variety of tasks from different areas, from object recognition to language modeling. This performance usually comes at a price: increasing empirical evidence, summarized today under the term "neural scaling laws" (Kaplan et al., 2020), suggests that model size controls a fundamental bound on performance, and this insight has recently driven a trend towards larger and larger neural networks. At the same time, neural networks see adoption in more and more applications, creating the need for parameter sharing platforms such as HuggingFace, where users can freely share parameters of their neural networks. Here, the large sizes of neural networks directly translate to operating costs for server storage and network traffic. Arguably even more importantly, the large file sizes of neural networks often makes it prohibitively impractical to deploy them on end-user devices. As a result, it is today the norm that applications relay any features that involve neural networks to a server, leading to increased latency and concerns regarding privacy and regulatory constraints (Saravanan & Kouzani, 2023; Hohman et al., 2024).

With this development, there has been rising interest in the development of data compression techniques for neural networks (Gholami et al., 2022; Marinó et al., 2023). Neural network compression can help with two problems: improving the *inference speed*, allowing for faster and more energy efficient neural networks (Chmiel et al., 2019), and decreasing the actual *network size*, reducing memory bottlenecks and hardware costs caused by storage and transmission of the model parameters. Most of the works we have surveyed focus on either one of these problems, but not both.

To improve the *inference speed* of neural networks, **quantization** (Jacob et al., 2017; Dettmers et al., 2022) aims to represent the weights of a neural network in a lower bit precision, for example 4 bit integers, which can be processed much faster than normal 16 or 32 bit precision floating point numbers on most GPUs. **Pruning** (LeCun et al., 1989; Blalock et al., 2020) tries to determine which of the weights in a neural network are the most important for the correct model output, and removes

a certain number of non-relevant weights, which allows for the usage of fast kernels for sparse matrix multiplication or even skipping computations completely when whole groups of neurons are removed in the case of structured pruning (He & Xiao, 2024). **Knowledge distillation** (Hinton et al., 2015; Gou et al., 2021) trains a small student network on the outputs of a larger teacher network, aiming to replicate the behavior of the larger teacher model exactly in the smaller model.

While these methods do reduce the required number or size of the parameters to store, their focus is not the size of the neural network in a compressed representation (as it would, e.g., be transmitted over the internet or stored in an end-user application). An important step to achieve a data representation of minimal size is **entropy coding** (Shannon, 1948; MacKay, 2003), where a probabilistic model of the data source is built and then used to encode more probable symbols to shorter bit-strings and less probable ones to longer bit-strings. Research on reducing the storage size of neural networks (Choi et al., 2020; Becking et al., 2023) is more scarce, and some techniques, such as quantizing to non-uniform grids using vector quantization (Baalen et al., 2024), prohibit inference speed-ups, such as GPU kernels that can operate in low-bit integer arithmetic.

In this work, we propose a method that combines advantages from both quantization and entropy coding, resulting in a practical and efficient algorithm. Our compression method

- achieves high compression strength (i.e., small compressed file sizes);
- is applicable to most neural networks without modifications;
- works in a post-training setting, i.e., no expensive re-training is required, and only a relatively small calibration data set is needed to estimate Hessians;
- allows for a smooth trade-off between compression strength and accuracy (and trying out many points on this trade-off is very cheap as no new Hessians have to be estimated);
- is compatible with existing methods for inference acceleration on GPUs through activation quantization, with a barely noticeable impact on model performance; and
- has very high decoding speed and sufficient encoding speed for large neural networks.

We term our method **OPTQ-RD**, as it generalizes the recent state-of-the-art quantization framework OPTQ (Frantar et al., 2023) by introducing a rate-distortion trade-off into the optimization objective. To estimate bit rates in this trade-off, we use the entropy model used by the DeepCABAC entropy coder (Wiedemann et al., 2020a), which is specialized to achieve high coding speeds and a high compression performance for neural networks. However, our method is agnostic to the exact entropy model used, which can be easily swapped out to fit the needs of a practitioner, who for example might opt to use a simpler model to achieve even higher coding speeds on heavily resource-constrained devices. We empirically verify the effectiveness of our algorithm on various neural network architectures from the computer vision community.

## 2 RELATED WORK

We make an effort to distinguish between methods focused on *inference speed* (quantization, pruning, knowledge distillation) and methods focused on *storage size* (entropy based methods, parameter sharing), although the techniques are often combined in some form.

Methods focused on inference speed can be categorized into methods that require (re-)training the neural network (such as knowledge distillation or quantization aware training (Baskin et al., 2021)) and post-training methods (such as post-training quantization and pruning), although the latter sometimes involve fine-tuning, a form of partial retraining. We focus on the post-training setting here, under which our method also falls under. *SynFlow* (Tanaka et al., 2020) prevents layer collapse in network pruning by using a data-independent score based on synaptic saliency. *SparseML* (Kurtz et al., 2020; Singh & Alistarh, 2020) provides an easy-to-use tool for inference speed-up on arbitrary networks using fisher information based pruning and structured sparsification. *PENNI* (Li et al., 2020) decomposes the filters of a convolutional neural network into a set of shared basis kernels, which are learned during a retraining process with a sparsity constraint.

On the frontier of storage-focused methods, efforts have been made to build very general compression algorithms and pipelines for neural networks. The *Neural Network Compression and Representation Standard* (Kirchhoffer et al., 2022) defines a compression pipeline encompassing parameter

reduction (pruning, sparsification, parameter sharing, et cetera), quantization and entropy coding. Additionally, the standard defines interoperability with well-known neural network exchange formats such as ONNX (Bai et al., 2019). *Universal Neural Network Compression* (Choi et al., 2020) uses universal quantization (Ziv, 1985), i.e., it randomly perturbs the weights of a neural network before applying vector quantization, allowing their compression scheme to work independently of the source statistics of the parameters. Wiedemann et al. (2020b) propose to use an entropy-constraint together with a sparsification process, resulting in parameter representations that allow for high compression ratios and while keeping the inference speed-ups from general pruning techniques.

Additionally, a large body of work has recently appeared on the compression of very large language models (Dettmers et al., 2022; Chee et al., 2024; Kim et al., 2024; Ding et al., 2024), but the proposed methods address issues specific to LLMs (e.g., activation outliers (Xiao et al., 2023; Lin et al., 2024)). By contrast, our proposed method is agnostic to the network architecture.

## 3 BACKGROUND

### 3.1 INFORMATION THEORY AND COMPRESSION

The goal of compression is to map data from a data source to short bit strings. *Lossy* compression further trades off inaccuracies in the reconstruction of the data for even shorter bit strings. Consider a data source $X \sim P_X, X \in \mathcal{X}$, a discrete reconstruction space $\hat{\mathcal{X}}$ and a distortion function $D \colon \mathcal{X} \times \hat{\mathcal{X}} \to [0, \infty]$. For a given acceptable amount $\mathcal{D}$ of expected distortion, lossy compression aims to find an encoder $e \colon \mathcal{X} \to \{0,1\}^* \coloneqq \bigcup_{k=0}^{\infty} \{0,1\}^k$ and a decoder $d \colon \{0,1\}^* \to \hat{\mathcal{X}}$ that minimize of the following *rate-distortion problem*[1] (where $|\cdot|$ denotes the length of a bit string):

$$RD(\mathcal{D}) = \min_{e,d} \mathbb{E}_{X \sim P_X}\big[|e(X)|\big] \quad \text{with the constraint} \quad \mathbb{E}_{X \sim P_X}\big[D(X, d(e(X)))\big] \leq \mathcal{D}. \quad (1)$$

To simplify Equation 1, we first break up the encoder $e$ into two steps: a (typically non-invertible) *quantization* step $q \colon \mathcal{X} \to \hat{\mathcal{X}}$, $q(X) = d(e(X))$, and an invertible *entropy-coding* step $b \colon \hat{\mathcal{X}} \to \{0,1\}^*$, $b(\hat{X}) = d^{-1}(\hat{X})$ (an optimal decoder $d$ is indeed invertible since reserving two different bit strings for the same reconstruction $\hat{X}$ would be wasteful). This separation allows us to easily estimate the length $|e(X)| = |b(q(X))|$ of the compressed representation via the source coding theorem (Shannon, 1948; MacKay, 2003). Assuming an optimal entropy coder $b$, this theorem states that

$$|b(\hat{X})| \in \big[-\log_2 P_{\hat{X}}(\hat{X}), -\log_2 P_{\hat{X}}(\hat{X}) + 1\big) \quad (2)$$

where $P_{\hat{X}}$ is the push-forward of $P_X$ along $q$. In practice, $P_X$ is usually not known, and so neither is $P_{\hat{X}}$, and one has to resort to an empirical model for $P_{\hat{X}}$ and optimize over its parameters.

For long bit strings (the relevant regime for data compression), the "+1" on the right-hand side of Equation 2 is negligible, and the bit rate under an encoder that is optimal for $P_{\hat{X}}$ can thus be accurately estimated by the *information content*, $|b(\hat{X})| \approx -\log_2 P_{\hat{X}}(\hat{X})$. Finally, if $RD(d)$ is assumed to be convex, we can enforce the distortion constraint in Equation 1 by a Lagrange multiplier $\lambda > 0$,

$$RD(\lambda) = \min_{q, P_{\hat{X}}} \mathbb{E}_{X \sim P_X}\big[D(X, q(X)) - \lambda \log_2 P_{\hat{X}}(q(X))\big]. \quad (3)$$

**Sub 1-bit bit rates.** Equation 2 fundamentally simplifies the rate-distortion optimization as it allows accurately estimating the bit rate $|e(X)| \approx -\log_2 P_{\hat{X}}(q(X))$ without having to explicitly construct an optimal entropy coder. It also allows splitting up the total bit rate for entropy-coding of a high-dimensional $\hat{X}$ into individual (amortized) bit rates for each vector component $\hat{X}_i$: assuming an autoregressive model $P_{\hat{X}}(\hat{X}) = \prod_{i=1}^{n} P_{\hat{X}}(\hat{X}_i \mid \hat{X}_{<i})$, where $n = \dim(\hat{\mathcal{X}})$, each component $i$ can be thought of as contributing $-\log_2 P_{\hat{X}}(\hat{X}_i \mid \hat{X}_{<i})$ bits to the total bit rate. Crucially, this split holds even if the individual amortized bit rates are below 1 bit per component, which would make them impossible to measure explicitly: to encode, e.g., $n = 1,000$ components with 0.3 bit of information content each, a practical near-optimal encoder such as arithmetic coding (Pasco, 1976; Rissanen & Langdon, 1979) would generate a bit string of length very close to $n \times 0.3 = 300$ bit.

---

[1]More general formulations admit for stochastic en-/decoders. But without an additional constraint such as realism, there always exists a pair of deterministic en-/decoders among the minimizers of Equation 1.

## 3.2 Entropy Coding: DeepCABAC

In Wiedemann et al. (2020a), the authors propose the compression method DeepCABAC, which is specifically designed to deal well with the common weight distributions of neural networks, which the authors found to be mostly symmetric, centered around zero and with quickly vanishing tails. The lossless entropy coder of DeepCABAC is based on the Context-based Adaptive Binary Arithmetic Coder (CABAC), used in the video codecs H.264/AVC (Marpe et al., 2003) and H.265/HEVC (Sze et al., 2014). CABAC contains a context model to adapt the coder on-the-fly to the statistics of the given data, making it universally usable. Additionally, stemming from its use in video coding, CABAC has a high throughput and allows very efficient encoding and decoding, making it suitable for encoding the large parameter sets from neural networks. In the rest of the paper, we use the term DeepCABAC to refer specifically to this entropy-coder.

In DeepCABAC, a quantized weight $\hat{w} \in \mathbb{Z}$ is binarized as follows: first, a series of flag bits are set, called *sigFlag* (whether the weight is 0), *signFlag* (whether the weight is positive) and *absGr(n)Flag* (whether the absolute value of the weight is $\geq n \in \{1, 2, 4, 8, \dots N\}$). If this this does not uniquely identify the weight, the remainder $\hat{w} - N$ is transmitted directly. Additionally, if the absolute value is larger than $N$, the remainder is transmitted using an Exponential-Golomb code. To increase the compression strength of the encoding scheme, a context model is used to predict the value of the flag bits. The context model is initialized to $0.5$, and every time it encounters a value of the flag bit, updates its probability model by a small step accordingly (increasing for 1, decreasing for 0). The context model is used as the probability model of an arithmetic coder (Pasco, 1976; Rissanen & Langdon, 1979; MacKay, 2003) and can thus shorten the expected length of the bit string even further, encoding probable flag bits with less than 1 bit on average. Since the context model is state-based and only depends on all previously seen values, it does not have to be transmitted to the decoder side as the decoder can reconstruct it step by step while decoding.

## 3.3 Quantization: OPTQ

When dealing with quantization, one has to choose a suitable distortion function $D$ in Equation 3. A naive way to define a distortion might be to measure the euclidean distance between the quantized and original weights $\|\boldsymbol{W} - \hat{\boldsymbol{W}}\|_2^2$. This corresponds to simple quantization methods such as round-to-nearest. However, closeness in weight space does not guarantee that the outputs of the network are close. The distortion that actually interests us is the model performance, for example classification accuracy or perplexity. This quantity is usually costly to evaluate and highly nontrivial to minimize. A suitable trade-off between these two extremes is the layer-wise loss (Nagel et al., 2020),

$$\mathcal{L}(\hat{\boldsymbol{W}}) = \|\hat{\boldsymbol{W}} \boldsymbol{X}_\ell - \boldsymbol{W}_\ell \boldsymbol{X}_\ell\|_2^2 \tag{4}$$

where $\boldsymbol{W}_\ell \in \mathbb{R}^{O \times I}$ are the weights of layer $\ell$ with $O$ output and $I$ input nodes, and $\boldsymbol{X}_\ell \in \mathbb{R}^{I \times B}$ are the inputs to layer $\ell$ resulting from a forward pass of a small set of $B$ calibration data points through the network. This loss has the advantage of having a very simple Hessian $\boldsymbol{H}_\ell \in \mathbb{R}^{OI \times OI}$ where the rows of $\boldsymbol{W}_\ell$ can be processed independently of each other:

$$\boldsymbol{H}_\ell = 2 \cdot \mathbf{1}_{O \times O} \otimes (\boldsymbol{X}_\ell \boldsymbol{X}_\ell^T) \tag{5}$$

where $\mathbf{1}$ is the identity matrix, and $\otimes$ denotes the Kronecker product. Equation 5 is expressed for general linear layers for notational simplicity, but its generalization to, e.g., convolutional layers is straight-forward. As the Hessian is block-diagonal with all blocks having the same value, from now on, we will use $\boldsymbol{H}$ to refer to a block $2\boldsymbol{X}\boldsymbol{X}^T \in \mathbb{R}^{I \times I}$, also dropping the subscript for the layer.

Based on this layer-wise loss, Frantar et al. (2023) propose OPTQ, a quantization method that can efficiently quantize networks of arbitrary sizes. It quantizes a neural network layer by layer. In each layer, the rows of the weight matrix $\boldsymbol{W}$ can be quantized in parallel. For each row $\boldsymbol{W}_{i,:}$, $i \in \{1, \dots, O\}$, OPTQ iterates over its components $W_{ij}$, $j \in \{1, \dots, I\}$. For each component, OPTQ first quantizes $W_{ij}$ and then optimally corrects the remaining weights $\boldsymbol{W}_{i,>j}$ in the row by minimizing $\mathcal{L}(\hat{\boldsymbol{W}})$, resulting in (Hassibi et al., 1993; Frantar & Alistarh, 2022; Frantar et al., 2023)

$$\boldsymbol{W}_{i,>j} \leftarrow \boldsymbol{W}_{i,>j} - \frac{W_{ij} - \hat{W}_{ij}}{\left[(\boldsymbol{H}_{\geq j, \geq j})^{-1}\right]_{jj}} \left[(\boldsymbol{H}_{\geq j, \geq j})^{-1}\right]_{j,>j} = \boldsymbol{W}_{i,>j} - \frac{W_{ij} - \hat{W}_{ij}}{C_{jj}} \boldsymbol{C}_{j,>j} \tag{6}$$

where $(\boldsymbol{H}_{\geq j, \geq j})^{-1}$ denotes the inverse of the lower right sub-block of $\boldsymbol{H}$ starting at row and column $j$, and the upper triangular matrix $\boldsymbol{C}$ is the transpose of the Cholesky decomposition of $\boldsymbol{H}^{-1}$.

---

**Algorithm 1** OPTQ-RD

---

**Require:** $\boldsymbol{W} = [W_{ij}]$          ▷ Input weight matrix of size rows $\times$ cols
**Require:** $\boldsymbol{H} = 2\boldsymbol{X}\boldsymbol{X}^T$       ▷ Hessian of layer-wise loss, see Equation 5; cols $\times$ cols
**Require:** $G = \{g_1, g_2, \ldots, g_m\}$       ▷ Quantization grid (e.g., as in Equation 8)
**Require:** $\lambda$       ▷ Rate-Distortion trade-off parameter
**Ensure:** $\hat{\boldsymbol{W}} = [\hat{W}_{ij}]$       ▷ Quantized output weight matrix
1: $\boldsymbol{C} \leftarrow \text{Cholesky}(\boldsymbol{H}^{-1})^T$       ▷ Upper triangular matrix with size cols $\times$ cols
2: $E \leftarrow \text{initEntropyModel}()$
3: **for** $j = 1$ to cols **do**
4:      **for** $i = 1$ to rows **do**
5:          Set $\hat{W}_{ij} \leftarrow Q_{\text{OPTQ-RD}}(W_{ij}, \lambda, S)$    ▷ See Equation 10; $S$ is the internal state of $E$.
6:          Update $\boldsymbol{W}_{i,>j} \leftarrow \boldsymbol{W}_{i,>j} - \frac{W_{ij} - \hat{W}_{ij}}{C_{jj}} \boldsymbol{C}_{j,>j}$       ▷ See Equation 6.
7:          $E.\text{update}(\hat{W}_{ij})$       ▷ Update internal state $S$ of the entropy model.
8:      **end for**
9: **end for**
10: **return** $\hat{\boldsymbol{W}}$

---

## 4 METHOD

We now present OPTQ-RD, our proposed compression method for neural networks. OPTQ-RD builds on the quantization method OPTQ (Frantar et al., 2023) (see section 3.3), but it introduces a rate-constraint into the quantization step so that the resulting quantized weights can be more effectively entropy coded by DeepCABAC (Wiedemann et al., 2020a) (see section 3.2).

The original OPTQ algorithm quantizes a given scalar weight $W_{ij}$ by simple rounding to the nearest grid point, using a uniformly spaced grid with standard absmax quantization Dettmers et al. (2022),

$$Q_{\text{absmax}}(W_{ij}) = \left\lceil \frac{m \cdot W_{ij}}{\|\boldsymbol{W}\|_\infty} \right\rfloor \cdot \frac{\|\boldsymbol{W}\|_\infty}{m} \tag{7}$$

where $\lceil \cdot \rfloor$ denotes rounding to integers, and $m \in \mathbb{N}$ controls the number of grid points. Thus, Equation 7 rounds by minimizing the $L_2$-norm,

$$Q_{\text{absmax}}(W_{ij}) = \arg\min_{g \in G}(W_{ij} - g)^2 \quad \text{where} \quad G = \left\{ (i/m) \cdot \|\boldsymbol{W}\|_\infty \mid i \in \{-m, \ldots, m\} \right\}. \tag{8}$$

OPTQ turns this simplistic optimization in weight space into an (approximate) optimization over a layer-wise loss function that takes second-order information into account by introducing a correction step for subsequent weights $\boldsymbol{W}_{i,>j}$ after quantizing each weight $W_{ij}$ (see Equation 6). However, standard OPTQ does take into account how much each quantized weight $\hat{W}_{ij}$ will contribute to the total bit rate when one entropy codes the quantized model to reduce its file size. We propose to take bit rates into account during the quantization step of OPTQ.

Algorithm 1 presents our proposed OPTQ-RD algorithm. It is analogous to the original OPTQ algorithm except for a different quantization method on line 5 and an extra model update on line 7, which we both discuss now. The proposed quantization step on line 5 uses a Lagrange multiplier $\lambda$ to optimize a trade-off between (i) the amount $\Delta_\mathcal{L}$ by which the layer-wise loss $\mathcal{L}$ (Equation 4) increases if we round the current weight $W_{ij}$ to $\hat{W}_{ij}$ and then correct subsequent weights $\boldsymbol{W}_{i,>j}$ using Equation 6, and (ii) the amount $R$ that $\hat{W}_{ij}$ contributes to the bit rate after entropy coding. By following the derivation from Hassibi et al. (1993) and substituting $w_q$ with $W_{ij} - \hat{W}_{ij}$, we find

$$\Delta_\mathcal{L} = \frac{1}{2} \frac{(W_{ij} - \hat{W}_{ij})^2}{[(\boldsymbol{H}_{\geq j, \geq j})^{-1}]_{jj}} = \frac{1}{2} \frac{(W_{ij} - \hat{W}_{ij})^2}{(C_{jj})^2}, \tag{9}$$

where $\boldsymbol{C}$ is the transpose of the Cholesky decomposition of $\boldsymbol{H}^{-1}$. To estimate the rate $R$ of each weight $\hat{W}_{ij}$, a simple method would be to use the information content $-\log_2 f(\hat{W}_{ij})$, where $f(g)$ is the empirical frequency of grid point $g \in G$ obtained by pre-quantizing the weights with a simple method such as round-to-nearest or vanilla OPTQ. However, we found that using the actual entropy

model that is used in a specialized entropy coding method such as DeepCABAC (see section 3.2) leads to much better results. DeepCABAC uses an autoregressive model $P_{\text{DeepCABAC}}(\hat{W}_{ij} \mid S)$, i.e., it has an internal state $S$ that needs to be updated after encoding each $\hat{W}_{ij}$ so it can adapt to the empirical distribution of quantized weights (see line 7 in Algorithm 1). Thus, we propose to use the following rate/distortion quantization method,

$$Q_{\text{OPTQ-RD}}(W_{ij}, \lambda, S) = \arg\min_{g \in G} \Delta_{\mathcal{L}} + \lambda R = \arg\min_{g \in G} \frac{(W_{ij} - g)^2}{2(C_{jj})^2} - \lambda \log_2 P_{\text{DeepCABAC}}(g \mid S) \quad (10)$$

where $G$ is the same grid as in Equation 8. For $\lambda = 0$, Equation 10 reduces to the original OPTQ (Equation 8). Runtime optimizations (e.g., grouping columns) are possible, cf. Frantar et al. (2023).

## 4.1 CHOOSING PER-LAYER COMPRESSION STRENGTH

In our method, $\lambda$ controls to how strongly we compress the network. Instead of using a uniform $\lambda$ for the whole network, we can also choose a different $\lambda_\ell$ for each layer $\ell$. We motivate one particular choice of selecting $\lambda_\ell$ in the following.

For vanilla OPTQ, the quantization procedure is invariant under independent scaling of each layer-wise Hessian $\boldsymbol{H}_\ell$, as this only changes the absolute value of the loss function in Equation 4, but not the optimal solution. This is no longer the case for OPTQ-RD, as we trade off distortion against rate. In fact, if we scaled the Hessians of different layers independently, $\boldsymbol{H}'_\ell = \boldsymbol{H}_\ell \cdot \alpha_\ell$, we would have to set $\lambda'_\ell = \frac{\lambda}{\alpha_\ell}$ for the solution of OPTQ-RD to remain unchanged. We notice this to be problematic with layers that include batch-norms, as these essentially scale the values of the calibration samples $\boldsymbol{X}$ flowing through the network. Therefore, we propose to use

$$\lambda_\ell = \lambda \cdot \text{Tr}(\boldsymbol{H}_\ell) \qquad (\text{i.e., } \alpha_\ell = 1/\text{Tr}(\boldsymbol{H}_\ell)) \tag{11}$$

for networks with batch-norms. Equation 11 renders the compression objective in Equation 10 invariant under scaling of the Hessian. We report this version of our method as *OPTQ-RD 1/tr*.

## 5 EXPERIMENTS

For our experiments, we evaluate the following networks: ResNet18, ResNet50 (He et al., 2016) (on CIFAR10), MobileNetV3 Large (Howard et al., 2019) and VGG16 (Simonyan & Zisserman, 2015) (on ImageNet). Additionally, we include the performance for ResNet34 in Appendix A.1 for space reasons. We use implementations from TorchVision for MobileNet and VGG16 and an implementation of user edaltocg of the ResNets on CIFAR10 from HuggingFace.

We implement our algorithm in PyTorch (Paszke et al., 2019) and do all computations either on an NVIDIA RTX 2080 (ResNets) or an NVIDIA A100 GPU (MobileNet, VGG16). To perform our algorithm, we first calculate the layer-wise Hessians (and the Cholesky decompositions of their respective inverses) with 40,000 calibration samples for ImageNet and 12,800 samples for CIFAR10, unfolding convolutional layers into linear ones. Then, we run Algorithm 1 for a set of different grid sizes $\{2, 4, 5, 7, 9, 16, 25, 36\}$ and $\lambda$ parameters (which are iteratively sampled to ensure that the curves have an accuracy resolution of $< 0.02$), resulting in a separate curve of model accuracy over bit rate for each grid size. As is typical in the literature of lossy compression, we refer to these as rate/distortion curves for short. To ease the presentation of our results, we only show the Pareto front of this set of rate/distortion curves, i.e., we iterate over a fine grid of values $\mathcal{A} = \{0.02, 0.04, \dots, 1.0\}$ for the model accuracy and report the lowest bit rate achievable over all recorded combinations of $\lambda$ and grid sizes where the model achieves at least the target accuracy (this corresponds to approximately solving Equation 1). For comparison, Appendix A.3, includes a graph where we draw all rate/distortion curves (sweeping over $\lambda$) for all included grid sizes. Runtimes are reported in Appendix A.4.

**Baselines.** For entropy coding, we use the DeepCABAC implementation from NNCodec (Becking et al., 2023), which we additionally report as a stand-alone baseline (*NNCodec*). Here, we scan the elements in row-major order and use the universal scalar quantization (URQ) option. To create the rate-distortion curve, we sweep over different values of the granularity of the grid (the qp parameter).

As additional baselines, we use two variants of vanilla OPTQ (Frantar et al., 2023). The first (*OPTQ+BZ2*) compresses the quantized weights from OPTQ with the well-known universal compression algorithm bzip2 (for optimal performance of bzip2, quantized weights are represented as bytes and concatenated without delimiters). This represents a typical approach on boosting the compression strength of a quantization method, used for example in Choi et al. (2020). The second variant (*OPTQ+DeepCABAC*) uses vanilla OPTQ for quantization and then DeepCABAC to compress the quantized weights. Additionally, as an ablation, we show the performance obtained by directly applying our rate-distortion quantization method without using the iterative weight-correction process from OPTQ (*Direct RD*). This corresponds to quantizing each weight with the objective

$$Q_{\text{Direct RD}}(W_{ij}, \lambda) = \underset{g \in G}{\arg\min}(W_{ij} - g)^2 \cdot H_{jj} - \lambda \log_2 P_{\text{DeepCABAC}}(g \mid S). \quad (12)$$

## 5.1 COMPRESSION PERFORMANCE

We report rate-distortion curves for our methods (*OPTQ-RD* and *OPTQ-RD 1/tr*) together with baselines in Figure 1. Our methods consistently perform better in terms of rate-distortion performance than all other tested methods. Additionally, we observe that scaling $\lambda$ with the trace of the Hessian (OPTQ-RD 1/tr) seems to be important for the performance of OPTQ-RD on ResNets, which use batch-norm layers that scale the layer outputs. On these nets, OPTQ-RD 1/tr clearly outperforms OPTQ-RD with uniform $\lambda$. In Table 1, we report the lowest bit rate achieved by each method while keeping 95% of the original performance of the network. There, our methods consistently achieve a >30% increase in compression strength (reduction of bit rate) over the baselines for all networks.

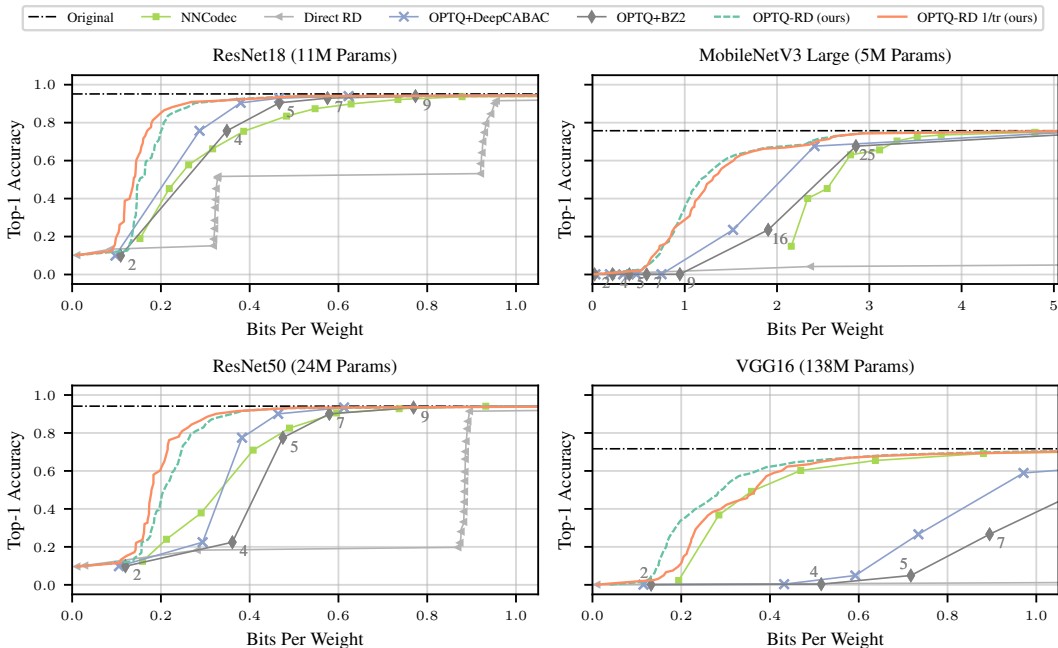

Figure 1: Rate-distortion curve for each tested method. Dotted black lines: original Top-1 accuracy of the network. ResNets were tested on CIFAR10; MobileNet and VGG16 on ImageNet. Small gray numbers next to OPTQ show the number of grid points that were used to construct the grid.

## 5.2 COMPATIBILITY WITH ACTIVATION QUANTIZATION

While our main focus is on reducing the cost of storage and transmission of neural networks, we also demonstrate that our method is compatible with activation quantization, which is the basis for inference acceleration on GPUs, see Nagel et al. (2021). We quantize VGG16 to the widely supported W8A8 (8-bit weights, 8-bit activations) format, using the PyTorch quantization library for activation quantization, and the quantized weights from the tested compression methods (where we typically use grids with far fewer than $2^8$ grid points anyway, see gray numbers in Figure 1). In

Table 1: Performance of different compression methods. For each method, we tried to find the best achievable compression performance (represented by the bits-per-weight), while retaining 95% of the original network accuracy. Best compression performance is marked bold. Compression Factor (CF) is reported assuming an original weight size of 32 bits-per-weight (BPW). Additionally, we report the compression factor and storage size obtained when accounting for the overheads required to store additional network information (refer to Appendix B.1 for more information).

| | Method | BPW ↓ | CF ↑ weights only | CF ↑ with overhead | Acc ↑ | Size with overhead |
|---|---|---|---|---|---|---|
| **ResNet18** | Direct RD | 0.96 | 33.47 | 31.67 | 0.92 | 1.4 MB |
| | NNCodec | 0.73 | 43.59 | 40.58 | 0.92 | 1.1 MB |
| | OPTQ+BZ2 | 0.47 | 68.56 | 61.36 | 0.90 | 728.5 KB |
| | OPTQ+DeepCABAC | 0.38 | 84.28 | 73.65 | 0.90 | 606.9 KB |
| | OPTQ-RD (ours) | 0.32 | 101.28 | 86.29 | 0.91 | 518.0 KB |
| | OPTQ-RD 1/tr (ours) | **0.27** | **118.92** | **98.76** | 0.91 | 452.6 KB |
| **ResNet50** | Direct RD | 0.89 | 35.77 | 30.85 | 0.90 | 3.0 MB |
| | NNCodec | 0.60 | 53.69 | 43.28 | 0.91 | 2.2 MB |
| | OPTQ+BZ2 | 0.58 | 55.22 | 44.26 | 0.90 | 2.1 MB |
| | OPTQ+DeepCABAC | 0.46 | 68.95 | 52.65 | 0.90 | 1.8 MB |
| | OPTQ-RD (ours) | 0.36 | 89.24 | 63.68 | 0.90 | 1.5 MB |
| | OPTQ-RD 1/tr (ours) | **0.32** | **98.53** | **68.26** | 0.90 | 1.4 MB |
| **MobileNetV3** | Direct RD | 5.40 | 5.93 | 5.52 | 0.72 | 4.0 MB |
| | NNCodec | 3.52 | 9.09 | 8.15 | 0.73 | 2.7 MB |
| | OPTQ+BZ2 | 5.85 | 5.47 | 5.12 | 0.76 | 4.3 MB |
| | OPTQ+DeepCABAC | 5.42 | 5.91 | 5.50 | 0.76 | 4.0 MB |
| | OPTQ-RD (ours) | **2.52** | **12.69** | **10.90** | 0.73 | 2.0 MB |
| | OPTQ-RD 1/tr (ours) | 2.62 | 12.20 | 10.53 | 0.73 | 2.1 MB |
| **VGG16** | Direct RD | 2.52 | 12.72 | 12.69 | 0.69 | 43.6 MB |
| | NNCodec | 0.88 | 36.33 | 36.08 | 0.69 | 15.3 MB |
| | OPTQ+BZ2 | 1.90 | 16.81 | 16.76 | 0.71 | 33.0 MB |
| | OPTQ+DeepCABAC | 1.61 | 19.87 | 19.80 | 0.71 | 28.0 MB |
| | OPTQ-RD (ours) | **0.65** | **49.17** | **48.71** | 0.68 | 11.4 MB |
| | OPTQ-RD 1/tr (ours) | 0.76 | 41.87 | 41.54 | 0.69 | 13.3 MB |

Figure 2, we see that activation quantization barely affects model accuracy, demonstrating that our weight compression method is also suitable for efficient inference. Note that we have not included NNCodec, as the quantized weights obtained from this method sometimes have more than $2^8$ points.

## 5.3 DETERMINING THE NECESSARY CALIBRATION SET SIZE

As we use a data-driven method to estimate the parameter sensitivity, we are naturally interested in how many samples are actually needed to achieve good results. In Figure 3, we have varied the amounts of samples used to estimate the Hessian of VGG16. We notice a saturation at around 40,000 samples. As ImageNet consists of 1,281,167 images, this corresponds to seeing a mere 3.12% of the training data just once, which is much cheaper than methods that rely on performing multiple full iterations over the training dataset during training or fine-tuning. Additionally, we also have to perform this iteration only once for the whole rate-distortion curve, allowing us to cheaply obtain versions of the network tuned for different performance levels (using different values for $\lambda$).

## 5.4 ENTROPY CONSTRAINTS CAN INDUCE SPARSITY

Entropy coding methods are often combined with some form of pruning (c.f. Choi et al., 2020) to achieve an even higher compression ratio. Additionally, weights with high sparsity ratios might enable further speed ups in inference. Instead of adding an explicit pruning step to our method, we observe that the weights we obtain tend to naturally be sparse for low bit-rates (i.e., high $\lambda$).

This is because the grid-value $g = 0$ usually incurs the lowest bit-cost (often by a large margin), as the weights of the neural network are approximately centered and symmetric around $0$, and Deep-CABAC assigns $0$ the lowest bit-cost by default (without taking into account the context-model). Figure 4 shows that OPTQ-RD indeed finds sparser weight matrices than NNCodec and plain OPTQ.

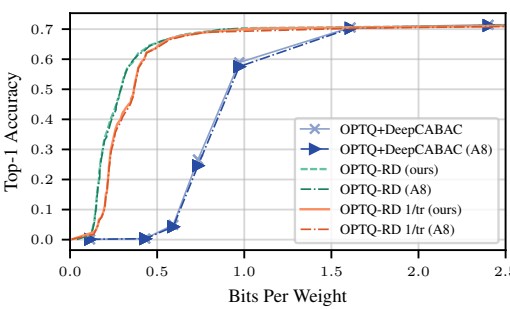 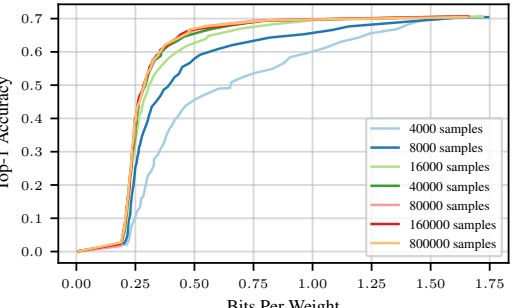

Figure 2: Performance of VGG16 before and after 8-bit activation quantization (A8).

Figure 3: Influence of the size of the calibration set on OPTQ-RD with uniform $\lambda$ on VGG16.

### 5.5 CALIBRATION SET AND TRAINING SET MISMATCH

Although the neural networks used in our experiments were trained on widely available datasets, in many real-world post-training settings one might not have access to the original training data. Therefore, we investigate the scenario where we calculate the Hessian using a different dataset than the one we use to evaluate the accuracy. We compress VGG16 using the Microsoft COCO dataset as a calibration dataset (Lin et al., 2014), and we evaluate on ImageNet. We use the same amount of samples as in the original VGG16 setup (40,000) and plot the rate-distortion curve for OPTQ-RD with uniform $\lambda$. In Figure 5, we can see that this causes a performance drop, although our method remains competitive with NNCodec while keeping its benefit of allowing faster inference.

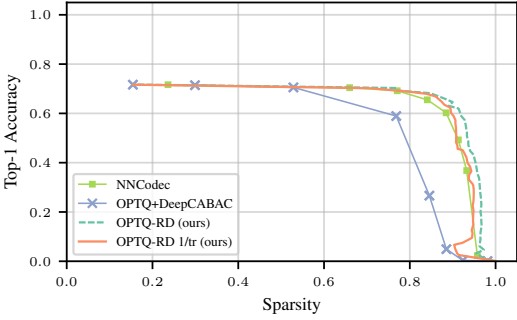 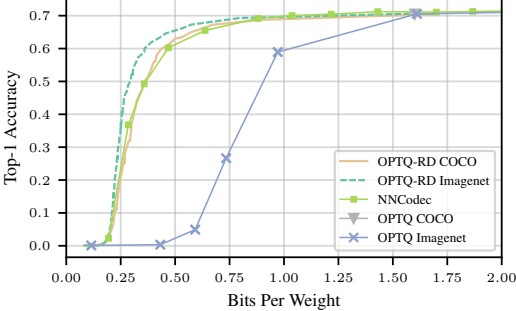

Figure 4: Sparsity of different compression methods in fraction of total weights $= 0$ (y-axis) against the classification accuracy (x-axis).

Figure 5: Effects of using a different calibration dataset than evaluation dataset (ImageNet) on OPTQ-RD with uniform $\lambda$ on VGG16.

## 6 CONCLUSION AND FURTHER WORK

In this paper, we proposed OPTQ-RD, a compression method that creates highly compressible quantized networks while keeping a simple and flexible uniform grid, suitable for accelerated inference. There are still some promising research directions not fully mapped out yet, as our method has multiple building blocks that can be experimented on. For example, one might explore using differently spaced grids to improve the compression performance even further (at the price of potentially slowing down inference), or use different entropy models than DeepCABAC. One might also try different methods of determining the compression strength $\lambda$ for each layer, for example by leveraging global second order information such as the fisher information content of the weights.

A harder problem would be to investigate rate-constrained compression for (very) large language models, as these are known to be much more difficult to compress down to the very low bit-rates

we have reported in our results. While the computer vision networks tested in our experiments can consistently be compressed to 0.5 bits-per-weight or lower with little performance drop, for LLMs, sparsification of more than 50% of the weights has only recently been achieved (Frantar & Alistarh, 2023) and even the sub 1-bit barrier has just been broken this year (Dong et al., 2024). These differences may indicate that language models may indeed encode more information per weight than (convolutional) computer vision models.

Additionally, we hope that this work inspires other research that combines more traditional ideas from general compression with modern techniques from the model compression community. For example, our observation that sparsification can result from an entropy constraint could provide some interesting avenue on works that combine quantization with pruning into a single, unified framework.

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

# A ADDITIONAL PLOTS

## A.1 RESULTS ON RESNET34

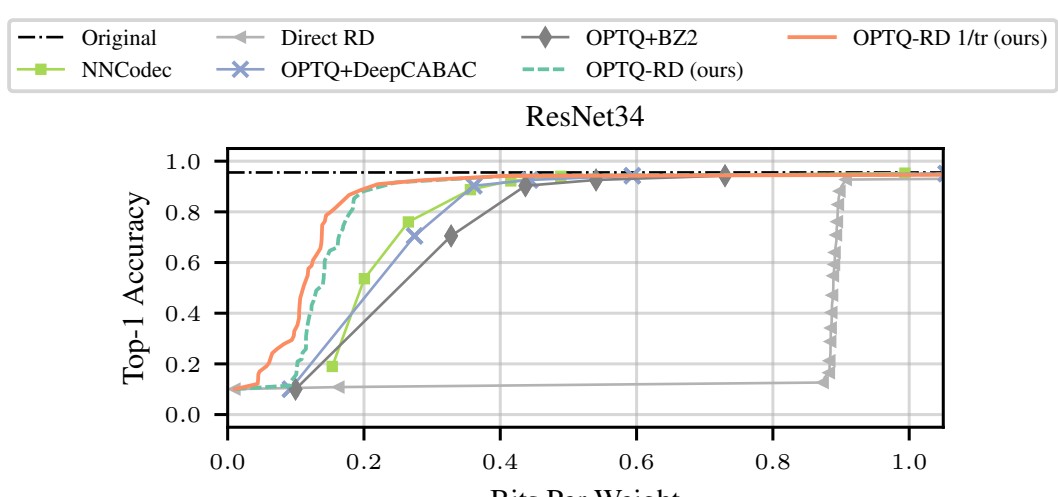

Figure 6: Performance of our methods on ResNet34.

## A.2 ROW-MAJOR VS COLUMN-MAJOR SCAN ORDER

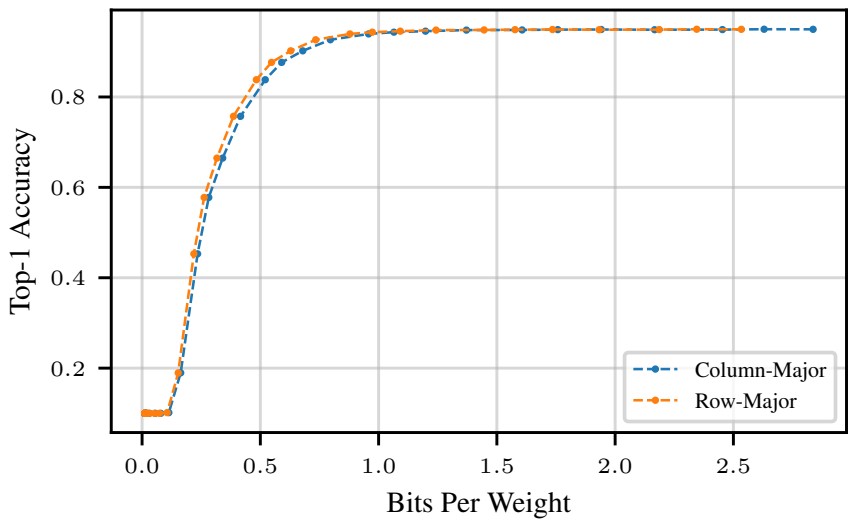

Figure 7: Effects of scan-order of weights when encoding ResNet18 with NNCodec.

## A.3 RATE-DISTORTION CURVES BY GRID SIZE

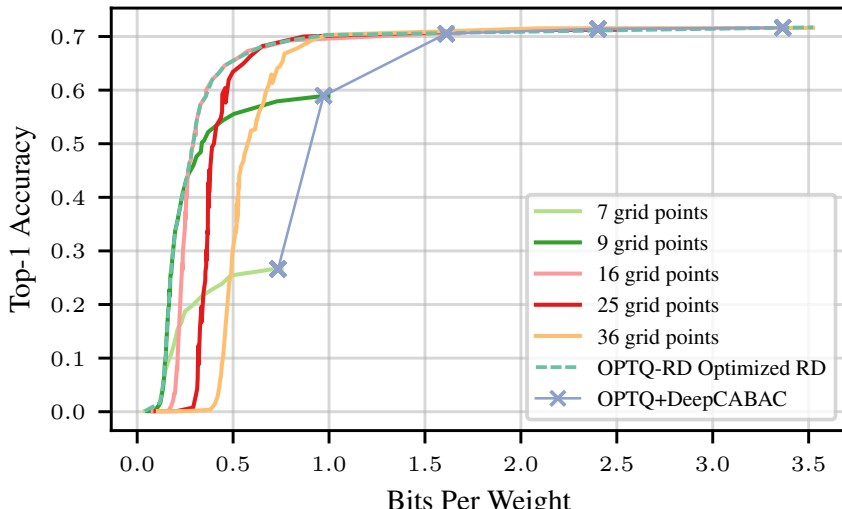

Figure 8: Performance of OPTQ-RD with uniform $\lambda$ on VGG16, shown for each tested grid size. The dashed curve corresponds to the RD-curve that we present on our other results, which is obtained by optimizing the bit-rate for different accuracy values over all possible grids.

## A.4 RUN-TIMES

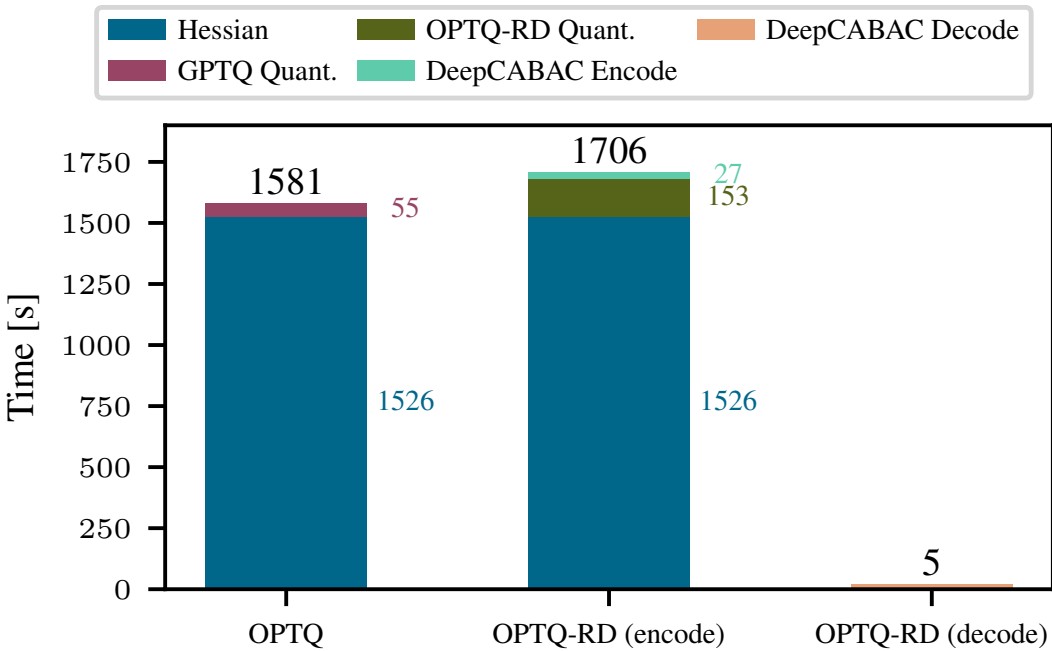

Figure 9: Run-times of our algorithm and OPTQ on VGG16 for a 16-point grid. We perform the hessian estimation (which includes calculation of the Cholesky) on an A100 GPU, the quantization and encoding/decoding are performed on a CPU. The hessian estimation only has to be done once for each network, subsequent runs for our method can use a saved hessian and then only incur the run-time for the quantization and DeepCABAC coding.

# B ADDITIONAL TABLES

## B.1 COMPRESSION OVERHEAD

To estimate the overhead for storing the compressed networks, we assume that batch-norms are not folded and all unquantised parameters (bias, batch norm statistics, scale factors $||\mathbf{W}||_\infty/m$ in Equation 7) are saved as fp32 (32 bits per parameter). This is the absolute *maximum* overhead needed, which can be reduced in practice through entropy coding, saving in fp16 or even lower, and by folding in batch norms into the preceding layers. Table 2 lists the resulting overheads.

Table 2: Actual storage size of the uncompressed neural networks (using 32 bits per parameters), together with the overhead needed for storage. The percentage indicates how large the overhead is in relation to the uncompressed network size.

| Network | Total Size | Overhead | Overhead (%) |
|---|---|---|---|
| ResNet18 | 44.7 MB | 77.0 KB | 0.17 % |
| ResNet34 | 85.1 MB | 136.6 KB | 0.16 % |
| ResNet50 | 94.1 MB | 425.5 KB | 0.45 % |
| MobileNetV3 Large | 21.9 MB | 294.6 KB | 1.34 % |
| VGG16 | 553.4 MB | 107.5 KB | 0.02 % |

