# OpenReview forum: "Rate/Distortion Constrained Model Quantization for Efficient Storage and Inference"
_ICLR.cc/2025/Conference — Submitted to ICLR 2025_

### Official Review · Reviewer_aZ9Q · 2024-10-28

**Soundness:** 3
**Presentation:** 2
**Contribution:** 2
**Rating:** 3
**Confidence:** 4

**Summary:**

The authors propose a method to combine compression with neural network quantization. The authors add a rate-distortion penalty to the rounding function and toggle its weighting to control the trade-off between compression rates and model quality. They conclude that their added compression penalty improves compression rates without severely impacting model quality.

**Strengths:**

The authors investigate combining ideas from the fields of neural network quantization and compression. They introduce a penalty to the rounding procedure that allows a practitioner to control the rate/distortion trade-off. The penalty works as they hypothesize.

**Weaknesses:**

While the experiments support their hypothesis, the baselines and models are insufficient. Since the authors are investigating post-training quantization techniques, it is expected to also test on language models. If model size is an issue for the authors, there are reasonably sized models to experiment with that are similar sizes to ResNet50 (e.g., Pythia-70M). Furthermore, if focusing only on convolutional architectures, it is important to test MobileNets, which are notoriously challenging and useful benchmarks. Finally, the design space for the Pareto frontier is unclear as the quantization levels for OPTQ are not disclosed.

**Questions:**

- What is the design space for the Pareto frontier?
- Did you quantize each of the models to different bit widths?
- If you are quantizing to 8-bit weights and activations before the penalty, it doesn't seem surprising that a compression penalty would cause significant savings. OPTQ (and other PTQ techniques for that matter) can push bit widths lower before significant degradation. Did you try quantizing weights and activations to 4 bits as the baseline?
- Why not test on parameter-efficient architectures such as MobileNet, EfficientNet, etc.?
- Why not test on small language models?

---

> ### Author Response · Authors · 2024-11-22
>
> We thank the reviewer for their feedback.
>
> > Since the authors are investigating post-training quantization techniques, it is expected to also test on language models.
>
> As we mention as an outlook in the conclusions, large language models have peculiar weight statistics, and pruning attempts in the literature have shown that these models behave very differently from traditional deep learning models in terms of where they store their information. To avoid publishing inconclusive and incomplete results regarding LLM compression, a thorough analysis of our method for LLMs would require a detailed analysis of how it interacts with the specific architecture, e.g.,
> - introducing independent $\lambda$ parameters for the different kinds of matrices (i.e., embeddings, MLPs, key, value, and query matrices); and
> - developing dedicated fast context models akin to DeepCABAC, which was not developed for the peculiar weight statistics of LLMs.
>
> This would have the effect that the bulk of our paper would focus on LLMs rather than the proposed compression algorithm. While we agree that neural network compression would be particularly desirable for LLMs, we consider it out of scope of this paper.
>
> We do not think that a new method has to be immediately demonstrated on the task that is known to be by far the most difficult. Our experiments span a wide range of model sizes and sparsities (thank you for the suggestion to also include MobileNets, see below). Our method _dramatically_ improves compression strength across all of them (consistently between 30% and 50%!). This clearly demonstrates that our method is widely applicable, and we think it would impede progress if results that show such dramatic improvements over a wide range of model sizes were held back from the community only because the initial proposal of a new method does not yet address an application domain that is well-known to be far more difficult.
>
> > [...] test MobileNets, which are notoriously challenging and useful benchmarks.
>
> Thank you for this suggestion! We added MobileNetV3 to Figure 1 and Table 1. Our method significantly outperforms all baselines on MobileNet, with similar margins as in the other networks.
>
> > What is the design space for the Pareto frontier?
>
> We describe the construction of the pareto front in the experiment section (lines 310-315). The set of considered grid sizes is specified on line 308 (2, 4, 5, 7, 9, 16, 25, and 36 grid points). Figure 8 in the appendix illustrates the construction of the pareto front. For a given rate, the Pareto front (dashed turquoise line) plots the highest achievable accuracy across all considered grid sizes (or, equivalently, the lowest achievable rate for any given accuracy).
>
> > Did you quantize each of the models to different bit widths?
>
> Each model was quantized to all of the grid sizes mentioned above (this is cheap to do because we can reuse the Hessians). For OPTQ+BZ2, each grid size corresponds to a single point in Figure 1 (which is annotated with a small gray number showing the grid size). For OPTQ-RD, each grid size corresponds to a section of the pareto front, see Figure 8.
>
> There might be a more general misunderstanding about the proposed method. The term "bit width" is only meaningful when one is concerned with holding the _uncompressed_ representation of a quantized weight in RAM (in which case one needs $\lceil \log_2|G| \rceil$ bits to represent an arbitrary point $g\in G$ from a grid of size $|G|$). In real _compression_ methods that employ an entropy coder (like our method and also OPTQ+BZ2), how the input data is represented (e.g., its "bit width") is irrelevant since a good entropy coder compresses it to the _information content_ ($-\log_2 P(g)$), which is bounded from above by $\log_2|G|$ in expectation (and, in our setups, is significantly lower, see bits-per-weight in Figure 1 and Table 1).
>
> > If you are quantizing to 8-bit weights and activations before the penalty, it doesn't seem surprising that a compression penalty would cause significant savings. OPTQ (and other PTQ techniques for that matter) can push bit widths lower before significant degradation. Did you try quantizing weights and activations to 4 bits as the baseline?
>
> We hope that the above clarifications resolve this question. As mentioned above and on line 308 of the paper, for both our method and the OPTQ+BZ2 baseline, we use grid sizes as low as 2 (i.e., a bit width of $\log_2 2=1$). We also include more realistic grid sizes, inculding 7 and 16 grid points (i.e., bit widths of 3 and 4, respectively), as well as more grid sizes between these values. Due to the use of an entropy coder, neither our method nor OPTQ+BZ2 is limited to integer bit widths, hence the inclusion of grid sizes that are not powers of 2 (powers of 2 play no special role in our method).
>
> We hope that our responses resolved the questions and clarified the distinction between our proposed compression method compared to pure quantization techniques.

---

> > ### Comment · Reviewer_aZ9Q · 2024-11-26
> >
> > Thank you for the MobileNetV3 results and the clarification on the bit rates.
> >
> > Respectfully, I would argue that your experiments do not span a wide range of model sizes as they are all within an order of magnitude of each other. Moreover, "wide applicability" is too strong a claim when the evaluations are focused on image classification models that are 5 years old at the youngest.
> >
> > While I agree that a new method does not have to be immediately demonstrated on the task that is known to be by far the most difficult, I would conversely argue that it should not be largely evaluated on tasks that are known to be by far the easiest (e.g., ResNets on CIFAR10). It is important to understand where the limits of your proposal are. The experimentation is insufficient to establish those limits. If you are unable to evaluate your technique on models and applications that are relevant to the community, then your result should at least be accompanied with a rigorous theoretical analysis of error limits, preferably with a proof.
> >
> > Furthermore, it is unclear to me why you expect LLM experiments to yield inconsistent results. You build your framework on top of OPTQ (formerly known as GPTQ), which is a post-training quantization method motivated by the challenges of LLMs. There have been several papers since OPTQ that have aimed at taming weight statistics (e.g., SmoothQuant, AWQ, QuIP, FrameQuant, etc.). Why are these techniques not complementary to your proposal, if weight statistics are the issue?

---

> > > ### Author Response · Authors · 2024-11-28
> > > **Author Response to Reviewer aZ9Q (1/2, Range of Experiments)**
> > >
> > > We thank the reviewer for their ongoing interest in our paper and their productive engagement in the discussion.
> > >
> > > We agree with the reviewer that it is always favorable to add more experiments for different architectures, tasks, and datasets. The generic nature of model compression makes this a daunting task. We don't make any claims about LLMs in our paper, and we elaborate below why we have deliberately chosen to focus on established computer vision architectures in our experiments, in line with recently published literature on quantization (without compression), which uses very similar ranges of models and data sets for their evaluations:
> > > - _[Sharpness-Aware Data Generation for Zero-shot Quantization](https://openreview.net/pdf?id=8mKXMnhnFW)_. ICML 2024. ResNets on CIFAR100, MobileNetV2 on ImageNet
> > > - _[Data-Free Quantization via Pseudo-label Filtering](https://openaccess.thecvf.com/content/CVPR2024/papers/Fan_Data-Free_Quantization_via_Pseudo-label_Filtering_CVPR_2024_paper.pdf)_. CVPR 2024. ResNets on CIFAR10/100, MobileNetV1 on ImageNet
> > > - _[MetaMix: Meta-State Precision Searcher for Mixed-Precision Activation Quantization](https://ojs.aaai.org/index.php/AAAI/article/view/29212)_. AAAI 2024. MobileNetV2, MobileNetV3, ResNet18 on ImageNet
> > >
> > >
> > > > I would conversely argue that it should not be largely evaluated on tasks that are known to be by far the easiest (e.g., ResNets on CIFAR10)
> > >
> > > We think describing our experiments as "by far the easiest" tasks would be a mischaracterization.
> > >
> > > First, criticising the simplicity of ResNets on CIFAR10 ignores the other tested networks. In particular, we now include experiments with a MobileNet trained on ImageNet, which the reviewer themself described as "notoriously challenging and useful benchmarks" in their original review. Please note that we did not have to change our method in any way when we performed this additional experiment – our method worked out of the box on this "notoriously challenging" benchmark and lead to a dramatic reduction in bit rate compared to all baselines (see Figure 1, top right panel).
> > >
> > > Second, we chose the other architectures (ResNets and VGG) _precisely because these are established architectures_ that are used in recent works on model quantization (see above). Also, the current compression standard for neural networks (MPEG-7 part 17) was developed mainly for such established architectures, and yet our method outperforms its implementation NNCodec significantly, for example by around 30% on ResNets, which is a massive gain by the standards of the compression community.

---

> ### Author Response · Authors · 2024-11-28
> **Author Response to Reviewer aZ9Q (2/2, Concerns about LLMs)**
>
> > Furthermore, it is unclear to me why you expect LLM experiments to yield inconsistent results. You build your framework on top of OPTQ (formerly known as GPTQ), which is a post-training quantization method motivated by the challenges of LLMs.
>
> We think there might still be a misunderstanding of our method: different to OPTQ, we propose a compression method, i.e., we care about information content, i.e., about the statistics of quantized weights. In this context, we kindly ask the reviewer to also take into account the discussion in our original response about "bit widths" and why they are not a meaningful concept in the context of entropy coding. We hope that this discussion helps clarifying the difference between quantization and compression by elaborating on why compression—in contrast to pure quantization—relies on an accurate entropy model, i.e., on a model of the distribution of quantized weights.
>
> Our method's dependency on an accurate entropy model is why we expect that experiments on LLMs would need to be quite elaborate and can't just be done as an additional point in the paper: LLMs are known to have very peculiar weight distributions, which, even though it does not affect OPTQ, will affect the requirements for our entropy model (and preliminary experiments on LLMs indeed indicate that DeepCABAC seems to be a poor entropy model for quantized LLM weights, but more specific claims would require a thorough evaluation and modeling of the statistics specifically of the various kinds of LLM weights such as token embeddings, MLPs, KVQ matrices, which we find is beyond the scope of a paper on general model compression).
>
> Additionally, we want to note that OPTQ is "motivated by the challenges of LLMs" only insofar as (quantized but uncompressed) GPU-memory and speed of execution (of the quantization) is concerned. OPTQ is a combination of speed-ups applied to the [Optimal Brain Compression](https://arxiv.org/abs/2208.11580) method, which is a general neural network compression algorithm and itself a small adaptation of [Optimal Brain Surgeon](https://doi.org/10.1109/ICNN.1993.298572) from 1992. OPTQ contains none of the "tricks" usually applied to LLM compression.
>
> > There have been several papers since OPTQ that have aimed at taming weight statistics (e.g., SmoothQuant, AWQ, QuIP, FrameQuant, etc.).
>
> We thank the reviewer for their suggestions. Please note that SmoothQuant is not suitable to help with weight compression, as it deals with _activation outliers_.
>
> **AWQ** proposes to scale the weights proportional to the magnitude of their activations, reducing the quantization error on them. As our method does not use Round-To-Nearest (RTN) quantization, but decides on weight quantization by a combination of distortion and bit-cost, it is not immediately clear how the proposed approach would interact with the quantization procedure. A possibility would be to add a loss term based on the activation strength to the optimization in $Q_{OPTQ-RD}$.
>
> **QuIP** proposes an algorithm that processes the Hessian and weight matrices with incoherence matrices, which enables them to use RTN to a great effect. QuIP changes the algorithm that OPTQ uses, removing the iterative weight updates (which turn out to be important in our setup, see our ablation _Direct RD_, which removes them). In particular, note that the theoretical analysis of QuIP requires the rounding to be _nearest-neighbour_, which is precisely the property that our R/D-quantization breaks (the R/D-quantization gets its entire performance gains from allowing us to quantize to grid points that are further away, provided that they are favorable on the rate/distortion trade-off).
>
> **FrameQuant** transforms the weights into a fusion-frame space through a set of projection matrices, and performs quantization in this noise-resistant space. Since FrameQuant produces quantized weights in a different space, we expect this transformation to significantly change the weight statistics and thus impact the achievable compression performance negatively, because our entropy model is tuned to the original weights statistics. Designing a corresponding fast entropy model in the transformed space and using it to combine our method and FrameQuant seems like an interesting research effort. Additionally, saving the required projection matrices would require additional storage, reducing the compression ratio, or incur a computational overhead for recalculating them each time the network is used.
>
> Overall, we think that all of these approaches sound like highly interesting research avenues. A thorough investigation that _compares different methods_ of simplifying the weight statistics of LLMs (including the ones mentioned above), analyzing their benefits and drawbacks, would produce a very relevant follow-up paper.
>
> Again, we sincerely thank the reviewer for making these invaluable suggestions and giving us pointers for interesting future research directions!

---

> ### Comment · Reviewer_aZ9Q · 2024-12-02
>
> Thank you for your responses. After careful consideration, I have decided to lower my recommendation. While the authors' ideas for combining post-training quantization with traditional compression techniques is promising, the study is incomplete and would benefit from more thorough experimentation and analyses.
>
> While ResNets can be useful benchmarks, CIFAR10 is no longer a relevant dataset for this setting. Furthermore, I also noticed that there is no comparison (or mention) of Deep Compression [1], a seminal work published in ICLR 2016. Deep Compression is 8 years old at this point and reported 49x compression rates on VGG16 on ImageNet. It is unclear if OPTQ-RD provides any significant uplift over this. Finally, as OPTQ was designed for and tested on LLMs, the author's acknowledgment of the challenges with LLMs implies a degradation in performance of that workload. This is an important challenge for the authors to either characterize or address in order for the work to be relevant to this conference.
>
> I emphasize that the authors' idea has potential and that there is a space for the intersection of traditional compression and quantization. **However, OPTQ was designed for language models and so any work building from this algorithm should highlight that performance is at the very least maintained.** Therefore, the lack of experimentation on language models is still concerning. The authors acknowledge that their technique may require a different entropy model, this feels necessary for this research to be impactful in this community.
>
>
> [1] Deep compression: Compressing deep neural networks with pruning, trained quantization and huffman coding, Han et al., ICLR 2016

---

> > ### Author Response · Authors · 2024-12-04
> > **Confusion about the sudden change of the rating**
> >
> > We appreciate the reviewer's continued engagement, but we are frankly having a very difficult time understanding the sudden change in their assessment. We find it unfortunate that the reviewer brought up two entirely new arguments only so briefly before the end of a 3-week discussion period.
> >
> > We think both new arguments are invalid: the paper cited in the reviewer's comment covers a very different problem than ours, and the discussion whether our algorithm maintains OPTQ's performance on LLMs disregards the fact that our algorithm _includes_ OPTQ as a special case (for $\lambda=0$), as explicitly mentioned in our paper.
> >
> > > Furthermore, I also noticed that there is no comparison (or mention) of Deep Compression [1], a seminal work published in ICLR 2016. Deep Compression is 8 years old at this point and reported 49x compression rates on VGG16 on ImageNet. It is unclear if OPTQ-RD provides any significant uplift over this.
> >
> > The paper the reviewer mentions proposes a compression method that re-trains the network, whereas our paper proposes a _post-training_ compression method. Compression-aware training is quite impractical because, in compression, one usually has to try out various quality settings ($\lambda$ for our method), which would require re-training the network several times. We highlight that our method achieves the _same compression ratio_ as the mentioned paper on VGG16, without requiring re-training. Additionally, our method is suitable for efficient inference (see new Figure 2), which the linked method is _not_, as it produces a *non-uniform-grid*. **Our method thus offers _huge_ practical advantages over the mentioned paper.**
> >
> > There is a large body of work that concerns different settings of parameter reduction in neural networks. It is important to exactly pin-point the use case, which in our case is a method suitable for _post-training_ quantization and compatible with _efficient inference_.
> >
> > > However, OPTQ was designed for language models and so any work building from this algorithm should highlight that performance is at the very least maintained.
> >
> > Trivially, our work maintains the performance of OPTQ, as our method is a _strict generalization_ of OPTQ, containing it as a special case: setting the $\lambda$ parameter to $0$ yields exactly the same result as running vanilla OPTQ, as then our quantization operation is identical to OPTQ. We explicitly mention this on lines 279-280 in our paper. We think that this makes it clear that our work always at least _maintains the performance_, as the reviewer has requested. **We stress that there can simply be no case where our method performs worse in storage than combining vanilla OPTQ with entropy coding.** We also stress that using an adaptable entropy coding model (which DeepCABAC falls under) practically always reduces the bit-size (compared to not using entropy coding) outside of near-adversarial examples. We have not encountered a single case where using DeepCABAC actually increased the storage demands when using it to entropy-code quantized network weights, also not on preliminary experiments on LLMs.
> >
> > ## Summary
> > We are quite confused about how this discussion thread has moved the reviewer to lower their score. In their initial review, the reviewer only suggested to add experiments on parameter-efficient networks (such as MobileNet), asked whether we explored low enough "bit widths", and why there were no experiments on LLMs. In our rebuttal, we added the requested MobileNet experiment (where our method significantly outperforms all baselines), explained that we already explore even more "bit widths" (really: grid sizes) than what the reviewer had suggested, and argued why we think that extending and evaluating our method to LLMs would be out of scope. The fact that we addressed these concerns (especially the additional experiments and the misunderstanding about "bit widths") was not acknowledged by the reviewer. Instead, the reviewer brought up completely new points less than two days before the end of a 3-week discussion period. These new points consist of a new reference that seems increasingly unrelated to our paper, and another question about our paper that was clearly already answered in the original submission.

---

### Official Review · Reviewer_rEH8 · 2024-11-03

**Soundness:** 3
**Presentation:** 3
**Contribution:** 3
**Rating:** 6
**Confidence:** 4

**Summary:**

This submission proposed to combine neural network quantization (after training, Post-Training Quantization (PTQ)) and parameters compression (compression) for storage in an end-to-end procedure. The quantized parameters are guided to compression-friendly distribution. Specially, after network is trained, it used OPTQ for parameter quantization. The OPTQ used is reformulated to combine DeepCABAC (a compression method) by Lagrange, leading to a quantization results considering compression requirement. The Lagrange factor trading off quantization and compression is layer-wise by considering the Hessian in each layer.

Experiments in Computer Vision tasks (ResNet series and ImageNet, CIFAR10) demonstrate its efficiency in performance and compression.

**Strengths:**

- The idea of the proposed method is straight forward and easy to understanding.
- It is interesting of combining quantization (for efficient inference) and compression (for storage).

**Weaknesses:**

- The main methods used in this submission is published and well-known. The submission merely combine the methods with a Lagrange formulation.

**Questions:**

- In experiments, the submission use Bits-Per-Weight to represents the compression performance, how is it related to the actual storage?
- As I understand, the compression take effects in disk (offline storage), instead of memory (online inference), is the proposed method benefit for inference speed?

**Details Of Ethics Concerns:**

N.A.

---

> ### Author Response · Authors · 2024-11-22
>
> Thank you for your review!
>
> > The main methods used in this submission is published and well-known. The submission merely combine the methods with a Lagrange formulation.
>
> We are frankly having difficulties following this line of reasoning. First, as far as we know, our method is the first post-training compression method of neural networks that optimizes global rate/distortion performance. The most closely related methods are OPTQ and NNCodec. OPTQ does not take bit rates into account (it is not a compression method), and NNCodec performs only scalar quantization (i.e., it quantizes each weight independently), which, as we show, empirically turns out to be a poor approximation. And despite addressing a more general problem, our method is compatible with activation quantization for inference speedup (see new Figure 2), which NNCodec is not (it needs a large quantization grid to obtain good performance).
>
> Second, our empirical results show a _dramatic_ improvement on an important task (see "Importance of weight compression" below) across all tested networks, which span a wide range of sizes. We believe the fact that such dramatic improvements can consistently be achieved with such a simple and well-motivated method makes it only more important to share these insights with a broad audience.
>
> Third, we agree that our method builds on existing ideas. Indeed, we would like to highlight that we bring together ideas from different communities: most works on model quantization in the machine learning (ICLR/NeurIPS/ICML) community have focused on computational efficiency. So far, there seems to have been little idea exchange with researchers from the signal processing (IEEE) community who focus on storage and bandwidth (e.g., with NNCodec/DeepCABAC).
>
> Finally, while the technique of Lagrange multipliers is standard, this is beside the point. At some level, most challenges in machine learning boil down to _identifying the relevant trade-offs_, and Lagrange multipliers are just the mathematical language of trade-offs. For an analogy, consider the beta-VAE paper (Higgins et al., ICLR 2017), which has been highly influential in the representation learning community. From a purely formal standpoint, the only proposed change in this paper is to introduce a Lagrange multiplier into a well-established loss function (even in front of an already established term!). Yet, the idea of considering the corresponding trade-off, and its thorough analysis, was absolutely crucial.
>
> > [relation between Bits-Per-Weight and actual storage]
>
> We have made the storage size clear by adding a column with the full file size of the entire compressed network to Table 1. This storage is the product of bits-per-weight and number of model parameters, plus a small overhead for storing scale factors, biases and batch-norms (see Appendix B.1 and Table 2). Note that this overhead always has the same size, independent of the methods used.
>
> > As I understand, the compression take effects in disk (offline storage), instead of memory (online inference), is the proposed method benefit for inference speed?
>
> This is correct, our work focuses on reducing the storage and transmission cost of neural networks (see motivation below). However, in contrast to the (few) existing post-training compression methods for this task like NNCodec, our method is also compatible with the common approach to speeding up inference by activation quantization. In fact, we find that activation quantization has an almost imperceptible effect on the accuracy of our compressed models (see newly added Figure 2), likely because we already quantize the weights to a fairly small grid anyway. We acknowledge that our claims regarding inference acceleration in the introduction were phrased confusingly, and we fixed the phrasing.
>
> **Importance of weight compression:** with its focus on storage size of neural networks, our paper addresses a problem that is not widely studied in the ML community. We acknowledge that this may make it harder to appreciate the importance of the problem. We kindly ask the reviewer to keep an open mind towards new problems because we believe that file sizes of neural networks are a severe issue with important consequences for society. Existing consumer applications of deep learning largely rely on cloud services, which are problematic in privacy sensitive domains (e.g., for lawyers, therapists, or investigative journalists), and which raise serious antitrust concerns. Due to these concerns, many recent works have focused on speeding up inference so that it can be done on consumer hardware. However, we argue that, even if inference on consumer hardware is fast, it will never catch on if it would mean that lots of apps on our phones had to permanently store hundreds of megabytes of neural network weights, even if we rarely use the app.
>
> We hope that our response and our additional experiments on activation quantization address all issues raised by the reviewer.

---

> > ### Comment · Reviewer_rEH8 · 2024-12-02
> > **I keep my score and stand for its acceptance.**
> >
> > Thanks for your response.
> >
> > First of all, the contribution (the first post-training compression method of neural networks that optimizes global rate/distortion performance), empirical improvemnt does not change the fact that the both parts of method is combined by Lagrange formulation. The pioneer, experimental results and formulation are seperated parts of the work.
> >
> > Besides, I don't think combination of exisiting works (such as the Higgins et al., ICLR 2017 you mention) is innovative.
> >
> > I want to highlight that: the combination of quantization and storage is interesting and pioneer, which is a point for acceptance. However, the formulation of using Lagrange is lacking of innovation, especially the main components are not original, which is a weak point for the innvation of the work.
> >
> > I stand for acceptance for the submission and I insist on my score for author does not provide evidences to address my concerns on innovation of formulation.

---

### Official Review · Reviewer_5jah · 2024-11-04

**Soundness:** 2
**Presentation:** 3
**Contribution:** 2
**Rating:** 5
**Confidence:** 3

**Summary:**

The paper presents OPTQ-RD, an extension of the OPTQ quantization method that introduces a rate-distortion trade-off. By adapting the quantization step to account for bit rate using an entropy model, the proposed framework compresses neural networks and maintains model accuracy. The method is experimentally evaluated on several computer vision models such as ResNets and VGG16 on CIFAR10 and ImageNet datasets.

**Strengths:**

1) The paper proposes a combination of quantization and compression methods, which is bridging the gap between optimizing for storage and inference. The integration of an entropy model (DeepCABAC) into the quantization process distinguishes the proposal from the other SOTA methods.
2) The proposal is evalauted on computer vision datasets to demonstrate its effectiveness.
3) The paper is well written and structured.

**Weaknesses:**

1) The generalizabity of the proposal on other model architectues and datasets is not evaluated explored. Addressing this issue running some extra experiments would strengthen the paper.
2) Table 1 presents the comparison with other compression techniques, however there is no comparison with the works reported in the related work section (e.g., those using quantization, pruning, and knowledge distillation). This could provide a more comprehensive performance context and help the reader understand better the benefits of the proposal.
3) It would be beneficial to mention what is the novelty of the proposal. The way it is written currently, it is seems that the proposal is incremental in terms of novelty as it is a combination of various existing techniques.

**Questions:**

1) How does the proposed method work for other architectures apart from VGG16 on ImageNet and the ResNets on CIFAR10?
2) What are the trade-offs in selecting the size of the calibration sets, and how does this affects models with significantly more parameters than the ones tested?
3) How does this method compares agaainst other SOTA works? Currently,  the proposal is compared with very limited SOTA works (Table 1)
4) Could you please mention the novelty of the proposal (please see the weaknesses above)?

---

> ### Author Response · Authors · 2024-11-22
>
> We thank the reviewer for their detailed comments.
>
> > - Weakness 1: [generalizability of the proposal on other model architectures and datasets]
> > - Question 1: [...] other architectures apart from VGG16 on ImageNet and the ResNets on CIFAR10?
>
> We added an experiment with MobileNetV3 (see updated Figure 1), finding similar results. Thus, we now analyze 5 networks that span a wide range of sizes and information content per weight (see varying x-axis scales in Figure 1) on 2 datasets, and we find that our method dramatically improves compression strength in all of them.
>
> > - Weakness 2: [...] no comparison with [...] quantization, pruning, and knowledge distillation
> > - Question 3: [...] compares against other SOTA works? Currently, the proposal is compared with very limited SOTA works (Table 1)
>
> Pruning and knowledge distillation are usually motivated by reducing computational and/or memory cost of inference. Our focus is on storage and transmission cost. This is the domain of source coding aka data compression.
>
> Data compression fundamentally requires quantization because one maps to the countable space of bit strings. Pruning and distillation are orthogonal to this: they can be combined with compression to reduce the number of weights that one has to compress, but if the goal is to reduce storage and transmission cost, then the remaining weights should still be compressed by any of the methods we evaluate (our compression factors of up to 100x would be unrealistic for pure pruning or distillation without compression).
>
> We could not find many other works on reduction of storage and transmission costs for neural network weights in a post-training setting. Apart from NNCodec, [universal compression](https://arxiv.org/abs/1802.02271) seems to be the only other competitor, but their reported compression ratio of 47.10 on ResNet-32 with CIFAR10 is much lower than the ~100x that we achieve on the ResNet family. We will add this as a remark to our paper (code for the universal compression paper seems to be unavailable, and given the large performance difference on ResNet, reimplementing it for evaluation on other models does not appear fruitful).
>
> > - Weakness 3: [...] mention what is the novelty of the proposal. The way it is written currently, it is seems that the proposal is incremental in terms of novelty as it is a combination of various existing techniques.
> > - Question 4: Could you please mention the novelty of the proposal [...]?
>
> Thank you for making us aware that this did not come across in our formulations. We will carefully clarify the phrasing.
>
> As far as we know, our method is the first post-training compression method of neural networks that truly optimizes global rate/distortion performance. The most closely related methods are OPTQ and NNCodec. OPTQ does not take bit rates into account at all (it is not a compression method), and NNCodec performs only scalar quantization (i.e., it quantizes each weight independently), which, as we show, empirically turns out to be a poor approximation. And despite addressing a more general problem, our method is compatible with activation quantization for inference speedup (see new Figure 2), which NNCodec is not (it needs very large quantization grids).
>
> These technical novelties are reflected in our empirical results, which we believe provide overwhelming evidence that the proposed method is anything but "incremental": we obtain a _dramatic_ improvement in compression strength across a wide range of model sizes and sparsities, consistently between about 30% and 50%. We agree that our method is simple, and we believe that it is very well motivated. This might make it seem almost obvious in hindsight. But we believe the fact that such dramatic improvements can consistently be achieved with such a simple and well-motivated method makes it only more important to share these insights with a broad audience.
>
> We agree that our method builds on existing ideas. Indeed, we would like to highlight that we bring together ideas from different communities: most works on model quantization in the machine learning (ICLR/NeurIPS/ICML) community have focused on computational efficiency. So far, there seems to have been little idea exchange with researchers from the signal processing (IEEE) community who focus on storage and bandwidth (e.g., with NNCodec and the DeepCABAC model).
>
> > - Question 2: [...] trade-offs in selecting the size of the calibration sets, and how does this affect models with significantly more parameters than the ones tested?
>
> We vary the calibration set size in Figure 3. There is little downside in choosing a larger calibration set if it is available, aside from a linear increase in calculation time for the hessians (which only needs to be done once per network for an entire R/D curve). Regarding dependency on network size: the [OPTQ paper](https://arxiv.org/abs/2210.17323) reports good results using only ~20 000 tokens for networks of up to 175B parameters.

---

> ### Author Response · Authors · 2024-12-02
> **Kind Reminder: Discussion period ends soon**
>
> As the discussion period is about to end very soon, we would like to kindly ask the reviewer to let us know if our above response addressed their concerns. For the final assessment of our paper, please take into account our additional experiments:
>
> 1. As requested by the reviewer, we have expanded our set of experiments by adding MobileNetV3 to the paper. We now evaluate 5 different networks of varying sizes total, on 2 datasets. Thus, we are in line with other published papers on parameter reduction techniques (see [our reply to reviewer aZ9Q](https://openreview.net/forum?id=LnKDcqOfgy&noteId=kRuFve5Z37)).
> 2. We have added an experiment on Activation Quantization, showcasing the practical applicability of our method.
>
> Additionally, we hope that our above response has clarified the difference of our work (focussing mainly on _compression_) compared to works such as pruning or knowledge distillation (focussing on parameter reduction or efficient inference). We hope that our response also clarifies why the amount of SOTA works we can compare to is relatively restricted (as model _compression_ is an underexplored field), and that it highlights the novelty of our approach, which combines ideas from different scientific communities.

---

### Official Review · Reviewer_awtQ · 2024-11-04

**Soundness:** 2
**Presentation:** 3
**Contribution:** 1
**Rating:** 6
**Confidence:** 5

**Summary:**

The paper proposes a post-training compression technique that combines recent advancements in quantization and compression. It extends the OPTQ framework by incorporating a rate-distortion trade-off, achieved through the addition of a bit-cost function inspired by NNCodec’s entropy model. The resulting OPTQ-RD method effectively balances compression strength and inference speed, achieving high compression ratios with minimal performance degradation across various CNNs.

**Strengths:**

The method can be applied without modifying network architectures, broadening its utility for different models.

The method’s ability to operate in a post-training setting with minimal calibration data makes it practical for real-world deployment.

The evaluation demonstrates that the proposed method consistently outperforms the baselines in terms of weight compression while preserving accuracy on par with them.

**Weaknesses:**

## Claims:

The paper does not reference previous methods that address the same problem [1,2].

The authors claim that OPTQ-RD achieves fast inference time; however, as I understand, only the weights are quantized, while activations remain in the floating point, typically resulting in a minimal reduction in inference time. This approach may reduce memory usage or storage, particularly in the context of compression and quantization for large language models (LLMs). However, it has a limited impact on convolutional neural networks (CNNs), especially those used in the experiments.


## Experiments:

The experimental setting is quite limited. VGG, which is not commonly used in practice, is known to have sparse weights, making it difficult to generalize the findings to more widely adopted network architectures.


Although bits per weight appears promising, Figure 1 and Table 1 provide little insight into its relevance to practical metrics, such as latency or energy consumption. Additionally, the plots suggest that, in intermediate cases, there is only a narrow range where the proposed method outperforms the vanilla baseline of OPTQ+DeepCABAC.


## Writing

The authors begin the abstract with a statement on pre-trained large models and discuss recent work on LLM quantization, yet these topics appear unrelated to the core of this study. Conducting evaluations only on CNNs without addressing LLMs or even ViTs is entirely valid; however, if they are not central to the paper, it raises the question of why these topics are mentioned at all.

### Refrences

[1]CAT: Compression-Aware Training for bandwidth reduction, Baskin et al., JMLR 2021

[2] Feature Map Transform Coding for Energy-Efficient CNN Inference, Chmiel et al. IJCNN 2020

**Questions:**

How does the choice of entropy model impact compression outcomes, and could other entropy models be seamlessly integrated into the method?

How does the method handle outliers in model weights, especially in architectures with significant variations in weight distributions? While this may seem out of scope, it could provide insight into the method's effectiveness in quantizing activations.

If the method is compatible with various quantization schemes, could you evaluate the impact of different schemes on large language models (LLMs)? A model with 1 billion parameters should be sufficient for this validation.

---

> ### Author Response · Authors · 2024-11-22
>
> We thank the reviewer for their constructive feedback. We are glad that they acknowledge that our method "consistently outperforms the baselines".
>
> > [...] reference previous methods that address the same problem [1,2].
>
> Thank you for pointing us to these interesting related works, which we added as references. Please note that [1] addresses compression during training rather than post training, and [2] addresses energy consumption during inference, which is very different from our focus on weight compression (except that our method is _compatible with_ activation quantization, see below).
>
> > VGG, which is not commonly used in practice, is known to have sparse weights, making it difficult to generalize the findings to more widely adopted network architectures.
>
> Aside from VGG, we also provide experimental results on three different ResNets, and we now added an experiment with a MobileNet to Figure 1. These 5 tested networks span a wide range of model sizes and information content per weight (see varying x-axis scales in Figure 1).
>
> > the plots suggest that, in intermediate cases, there is only a narrow range where the proposed method outperforms the vanilla baseline of OPTQ+DeepCABAC.
>
> We are having difficulties understanding this comment. Could you please clarify in which sense the improvements are limited to a narrow range?
>
> Looking at Figure 1, we find that our method outperforms OPTQ+DeepCABAC dramatically (between 30% to 50% rate reduction) for _almost all target accuracies_ in all tested networks. The only exceptions are the two limits where (a) accuracy converges to random guessing (an irrelevant regime for practical applications) and (b) the lossless-compression limit (where we doubt that any method can do significantly better).
>
> > [...] claim that OPTQ-RD achieves fast inference time; however, [...] only the weights are quantized [...]
> > little insight into [...] latency or energy consumption
>
> You are right, our claim regarding inference speedup was phrased confusingly. We clarified it and added new experimental results to substantiate it.
>
> The main contribution of our paper is a highly effective post-training compression method for neural networks that achieves dramatically smaller file sizes (i.e., bit rates) than prior work. We believe that this is important to make inference on end-user devices practical without requiring users to store hundreds of megabytes of network weights even for apps they might rarely use.
>
> At the same time, we acknowledge that a compressed model is only useful on consumer hardware if inference in it can be done fast. This is why we highlight that our method is compatible with activation quantization (in contrast to, e.g., NNCodec, which requires large quantization grids). We added Figure 2, which shows that 8-bit activation quantization indeed barely affects the performance of our compressed models.
>
> > [...] Conducting evaluations only on CNNs without addressing LLMs or even ViTs is entirely valid; however, if they are not central to the paper, it raises the question of why these topics are mentioned at all.
>
> We agree and shortened the differentiation of our work to LLM compression in the "Related Work" section. We kept the discussion in the conclusions, though, since this is meant as an outlook to possible future work.
>
> > How does the choice of entropy model impact compression outcomes, and could other entropy models be seamlessly integrated into the method?
>
> We briefly mentioned this on lines 85-88. Integrating other entropy models is straight-forward in principle, however, DeepCABAC was carefully designed to be very fast (by using lookup tables). We briefly mention on lines 266-270 that we initially tried a much simpler (stateless) entropy model using empirical frequencies of a prior OPTQ run (followed by entropy coding with either this model or DeepCABAC), but using DeepCABAC already during quantization worked much better.
>
> > How does the method handle outliers in model weights [...]? While this may seem out of scope, it could provide insight into the method's effectiveness in quantizing activations.
>
> We did not find outliers to be a problem in the networks we tested, but our method is compatible with preprocessing methods such as [weight equalization](https://arxiv.org/abs/1906.04721). Choosing a channel-wise grid might also help combatting outliers (our grids are layer-wise). Regarding activation quantization, we added Figure 2 (see above).
>
> > [...] impact of different [quantization] schemes on LLMs
>
> This would be very interesting, but also a very large field, which is why we leave it as future work to explore, e.g.,
> - using separate $\lambda$s for the embeddings, MLPs, key, value, and query matrices;
> - developing fast context models (akin to DeepCABAC) specialized to the peculiar weight statistics of LLMs.
>
> Thank you again for your detailed review. We hope that our response and additional experiments (activation quantization) resolved your concerns.

---

> ### Comment · Reviewer_awtQ · 2024-11-23
>
> Dear Authors,
> Thank you for your detailed response and the additional experiments. Your explanations have fully addressed my concerns, and I am pleased to increase my score to 6.

---

### Meta-Review · Area_Chair_spBC · 2024-12-24

**Metareview:**

There seems to be concerns regarding the novelty of the proposed approach and the extend to which the experiments are performed to showcase its generalizability (all reviewers).

The approach seems to be a simple extension of OPTQ. _I am not against simple extensions at all_. In fact, I like them the most when are shown to be effective on a wide range of scenarios which is lacking in this work at the moment.

The main motivation behind this work is the urgent need to compress giant large models into smaller file sizes to reduce the space (and transfer time) that's needed to store them in low-cost devices. Combining such methods with activation quantization has the potential to reduce the inference cost as well. However, the experiments in this work are performed primarily on small scale models (VGG, ResNet, MobileNet) and small datasets which doesn't align well with the presented motivation. This has been one of the major concerns across all the reviewers and I agree with them.

Showing the effectiveness of the proposed method in recent models is crucial. In fact, OPTQ, on which this paper is mostly based on, has been shown to quantize GPT models. I understand that showing results on language models might require additional work (e.g., developing models similar to DeepCABAC, as pointed by the authors), however, I think such contributions will be necessary to add novelty and show generality of the proposed method. And the fact that it's a post-processing method, I'd imagine it to be computationally less demanding.

I'd strongly suggest the authors to incorporate the thoughtful comments by the reviewers to increase the visibility of this work.

**Additional Comments On Reviewer Discussion:**

- The major concerns raised during the rebuttal were related to the lack of novelty and the lack of effective experiments (please see my comments above).
- Authors provided arguments behind why they could not perform experiments similar to QPTO (method on which this paper is primarily based on), however, these arguments also raise concerns regarding the applicability of the method.
- I appreciate the engagement the authors showed during the rebuttal, unfortunately the rebuttal did not provide convincing arguments for a clear acceptance of this work in its current form.

---

### Decision · Program_Chairs · 2025-01-22

Reject